# Loss of the m6A methyltransferase METTL3 in monocyte-derived macrophages ameliorates Alzheimer's disease pathology in mice

Huilong Yin[1,2,3�) ], Zhuan Ju[1☉], Minhua Zheng[4], Xiang Zhang[3], Wenjie Zuo[1,5], Yidi Wang[6], Xiaochen Ding[7], Xiaofang Zhang[3], Yingran Peng[3], Jiadi Li[1], Angang Yang[1,2,3]*, Rui Zhang[2,3]*

1 Henan Key Laboratory of Immunology and Targeted Therapy, School of Laboratory Medicine, Xinxiang Medical University, Xinxiang, Henan, China, 2 The State Key Laboratory of Cancer Biology, Department of Immunology, Fourth Military Medical University, Xi'an, Shaanxi, China, 3 The State Key Laboratory of Cancer Biology, Department of Biochemistry and Molecular Biology, Fourth Military Medical University, Xi'an, Shaanxi, China, 4 The State Key Laboratory of Cancer Biology, Department of Medical Genetics and Developmental Biology, Fourth Military Medical University, Xi'an, Shaanxi, China, 5 Xinxiang Key Laboratory of Tumor Microenvironment and Immunotherapy, School of Laboratory Medicine, Xinxiang Medical University, Xinxiang, Henan, China, 6 Department of Thyroid, Breast and Vascular Surgery, Xijing Hospital, Fourth Military Medical University, Xi'an, Shaanxi, China, 7 Department of Experimental Surgery, Xijing Hospital, Fourth Military Medical University, Xi'an, Shaanxi, China

☉ These authors contributed equally to this work.
* agyang@fmmu.edu.cn (AY); ruizhang@fmmu.edu.cn (RZ)

**Data Availability Statement:** All relevant data are within the paper and its Supporting information files.

## Abstract

Alzheimer's disease (AD) is a heterogeneous disease with complex clinicopathological characteristics. To date, the role of m6A RNA methylation in monocyte-derived macrophages involved in the progression of AD is unknown. In our study, we found that methyltransferase-like 3 (METTL3) deficiency in monocyte-derived macrophages improved cognitive function in an amyloid beta (Aβ)-induced AD mouse model. The mechanistic study showed that that METTL3 ablation attenuated the m6A modification in DNA methyltransferase 3A (*Dnmt3a*) mRNAs and consequently impaired YTH N6-methyladenosine RNA binding protein 1 (YTHDF1)-mediated translation of DNMT3A. We identified that DNMT3A bound to the promoter region of alpha-tubulin acetyltransferase 1 (*Atat1*) and maintained its expression. METTL3 depletion resulted in the down-regulation of ATAT1, reduced acetylation of α-tubulin and subsequently enhanced migration of monocyte-derived macrophages and Aβ clearance, which led to the alleviated symptoms of AD. Collectively, our findings demonstrate that m6A methylation could be a promising target for the treatment of AD in the future.

## Introduction

Alzheimer's disease (AD), as an age-related neurodegenerative disease, is the most common cause of dementia [1]. Among the common neuropathological features in AD are synaptic and neuronal dysfunction, intracellular neurofibrillary tangles, elevated levels of toxic amyloid beta

**Funding:** This study was supported by grants from the National Natural Science Foundation of China (31801128 to Y.H.L., 81630069, 31771439 to Y.A., 82173046 to Z.R., 82173162 to Z.X.), the Program for Ph.D. Starting Research Funding from Xinxiang Medical University grant 505249 to Y.H.L., and the National Key Research and Development Program grant 2016YFC1303200 to Z.R.. The funders had no role in the study design, data collection and analysis, decision to publish, or preparation of the manuscript.

**Competing interests:** The authors have declared that no competing interests exist.

**Abbreviations:** AD, Alzheimer's disease; Aβ, amyloid beta; BMDM, bone marrow-derived macrophage; ChIP, chromatin immunoprecipitation; CNS, central nervous system; DMEM, Dulbecco's Modified Eagle's Medium; KO, knockout; M-CSF, macrophage colony-stimulating factor; mRNA, messenger RNA; MWM, Morris water maze; qRT-PCR, quantitative reverse transcription PCR; WT, wild type; RIP-qPCR, RNA immunoprecipitation-qPCR; YMT, Y maze test.

(Aβ), and extracellular Aβ deposition as neuritic plaques [2–4], which induce cognitive decline. Consequently, there has been tremendous interest in elucidating the molecular mechanism underlying the pathological process of AD.

In recent years, chemical modifications in RNA have been recognized as important mechanisms for the regulation of gene expression and protein translation [5,6]. N6-methyladenosine (m6A), the most diverse and reversible posttranscriptional modification of eukaryotic messenger RNAs (mRNAs), is a strong regulator of mRNA splicing, stability, localization, and translation [5,7], which depends on the combined activity of methyltransferases and demethylases. Currently, the known methyltransferase complex is mainly composed of methyltransferase-like protein 3 (METTL3), methyltransferase-like protein 14 (METTL14), and Wilms tumor 1-associating protein (WTAP), while demethylases include obesity-associated protein (FTO) and AlkB homolog 5 (ALKBH5) [8–11]. Recent studies have demonstrated that m6A is involved in the development of the nervous system and neural degenerative diseases [12–14]. The widespread presence of m6A in the neuronal transcriptome also suggests its various functional roles in brain development and function [15–17]. In addition, accumulating evidence has shown essential roles of m6A modification in learning and memory through regulation of the translation of plasticity-related genes in the mouse brain [13,18–21].

Recently, it has been reported that blood-derived myeloid cells can cross the blood–brain barrier and differentiate into fully functional macrophages [22–24]. Blood-derived myeloid cells, recruited by central nervous system (CNS) damage, are considered microglial reinforcements of comparable functions and are accordingly termed "blood-derived macrophages" [25–27]. Activated microglia and blood-derived macrophages, often collectively referred to as CNS macrophages, adopt a variety of functional phenotypes that contribute to the progression of neurodegeneration as well as CNS repair and protection [28,29]. However, the roles of monocyte-derived macrophages in the development of AD remain to be fully elucidated.

In this study, we aimed to elucidate the role of m6A mRNA methylation in AD progression by conditionally inactivating the *Mettl3* gene specifically in myeloid cells using a *Mettl3* conditional mouse line in combination with *Lyz2*-Cre driver lines. We found that METTL3 ablation enhances the infiltration of monocyte-derived macrophages in an Aβ-induced AD mouse model. Further analysis showed that METTL3 depletion impairs the m6A modification in *Dnmt3a* mRNAs, which in turn attenuates the translation of DNMT3A. DNMT3A binds to the promoter region of *Atat1* and maintains its expression. Loss of METTL3 results in reduced α-tubulin acetylation and enhanced monocyte-derived macrophage migration, Aβ clearance and relief of AD symptoms.

## Results

### METTL3 deficiency in monocyte-derived macrophages attenuates the symptoms of Aβ-induced AD

Emerging studies have shown that m6A, the most abundant modification in eukaryotic RNA, plays a critical role in various developmental processes. However, the role of m6A RNA methylation in AD is unclear. To explore the involvement of monocyte-derived macrophages regulated by m6A during AD progression, we first showed that METTL3 did not alter the expression of surface cell markers (CD11b, F4/80) on macrophages from WT (*Mettl3*^fl/fl^*Lyz2*^-/-^) and KO (*Mettl3*^fl/fl^*Lyz2*^Cre/-^) mice (S1A Fig). We then examined whether METTL3 ablation in myeloid cells affects cognition in an Aβ-induced AD mouse model. We assessed the cognitive function of WT and KO mice by the Morris water maze (MWM) task and Y maze test (YMT). The results demonstrated that the learning and memory of the KO mice were improved compared with those of the WT mice (Fig 1A–1H). We also assessed

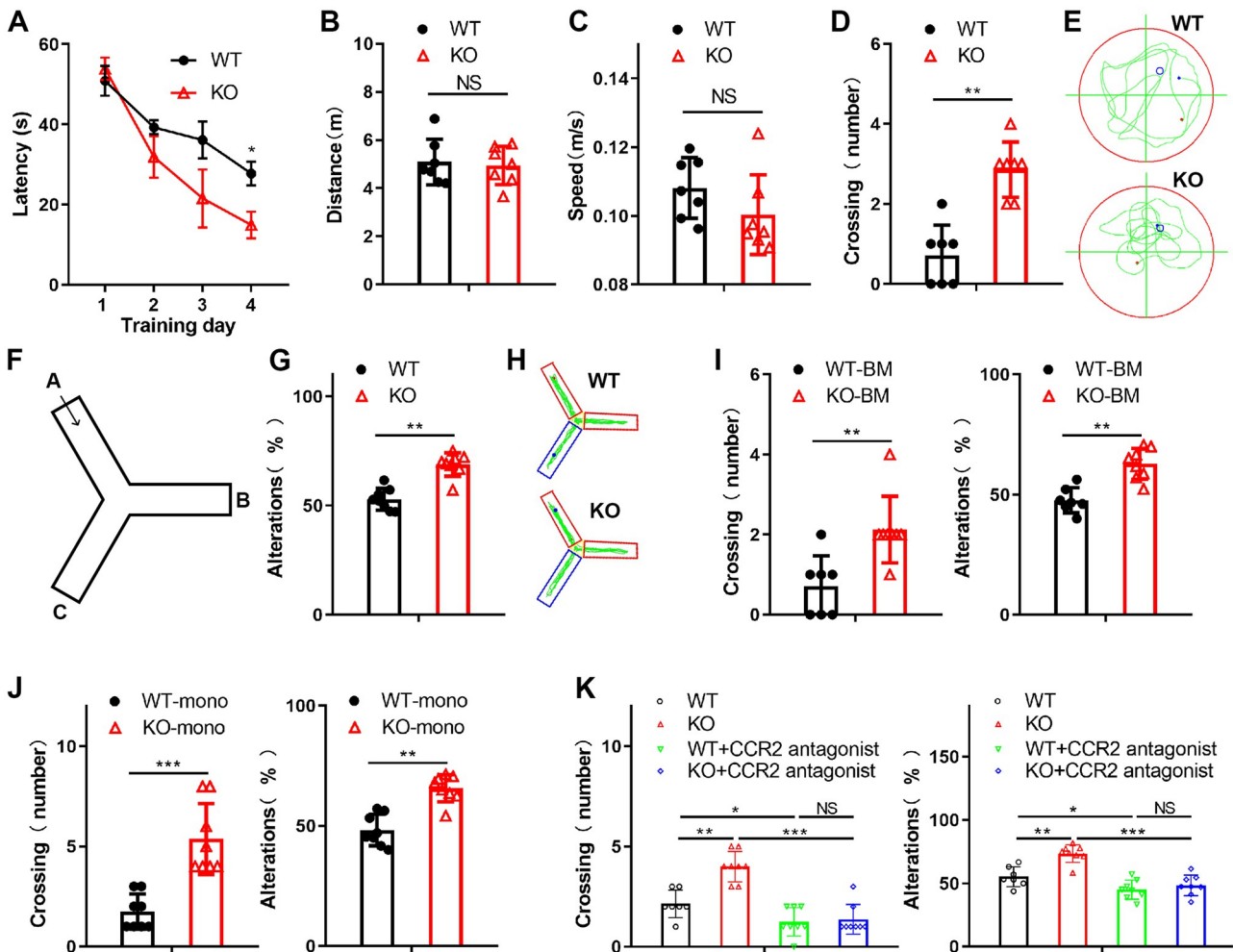

**Fig 1. METTL3 deficiency in monocyte-derived macrophage improve the cognition in a mouse model of AD. (A–E)** The WT or KO mice were trained and learned to find a hidden platform over 4 consecutive days. The escape latencies of the WT or KO mice in the MWM task on training days were recorded (A). On Day 5, probe trials were performed with the platform absent, and the travel distance (B), swim speed (C), and number of platform area crossings (D) were assessed. **(E)** Representative MWM swim plots for the WT and KO mice on Day 5. (F) Experimental design of the YMT. (G) The spatial working memory of the WT or KO mice induced by Aβ in the YMT. Alternations were counted as the percentage of "correct" alternation/total entries. "Correct" alternation means entry into all 3 arms on consecutive choices. (H) Representative YMT plots for the WT and KO mice induced by Aβ. (I) The learning and memory impairment of the Aβ-treated WT mice reconstituted with either WT or KO BM was examined by the MWM task and YMT. (J) The spatial learning and memory of mice with WT or KO monocyte transplantation were examined by the MWM task and YMT. (K) The learning and memory impairment of the Aβ-treated WT or KO mice injected with the CCR2 antagonist PF-4136309 (2 mg/kg) was assessed by the MWM task and YMT. All data in the figure are shown as the mean ± SD except for (A). $P < 0.05$ (*), $P < 0.01$ (**), $P < 0.001$ (***), NS means no significant difference. Underlying data can be found in S1 Data. Aβ, amyloid beta; AD, Alzheimer's disease; BM, bone marrow; KO, knockout; MWM, Morris water maze; WT, wild type; YMT, Y maze test.

the locomotor and exploratory behavior of mice by the open-field test. No significant difference in rotarod performance was observed between the 2 groups, indicating that the KO mice did not have compromised motor function (S1B Fig). Moreover, we generated chimeric mice by subjecting WT mice to bone marrow transplantation from WT or KO mice. Then, the cognitive function of the mice was tested by the MWM task and YMT. The results suggested that METTL3 deficiency ameliorated learning and memory deficits induced by Aβ in mice (Fig 1I). We also transferred monocytes isolated from the bone marrow of WT or KO mice into WT mice. The MWM task and YMT showed that the mice with METTL3-deficient monocyte transfer displayed ameliorated symptoms of Aβ-induced AD (Figs 1J and S1C).

Furthermore, we observed that inhibition of the recruitment of monocyte-derived macrophages into the brain ablated the effect of METTL3 deficiency on AD progression (Figs 1K and S1D). These results suggest that METTL3 depletion in monocyte-derived macrophages plays an important role in AD progression.

## METTL3 deficiency in BMDMs attenuates microtubule acetylation through ATAT1

A previous study reported that α-tubulin acetylation is closely related to neurodegenerative disease [30]. To investigate the role of α-tubulin acetylation in monocyte-derived macrophages involved in AD progression, we first examined the acetylation of microtubules in bone marrow-derived macrophages (BMDMs). As shown in Fig 2A, the level of microtubule acetylation was attenuated significantly in KO BMDMs compared with WT BMDMs. Similar results were also obtained in METTL3 knockdown BMDMs (Fig 2B). Furthermore, we detected the expression of tubulin acetyltransferase (*Atat1*) [31] and histone deacetylases (*Hdac6*) [32] and sirtuin type 2 (*Sirt2*) [33]. The results demonstrated that the expression of the major tubulin acetyltransferase *Atat1* was noticeably reduced in the KO BMDMs, while the expression of the major tubulin deacetylases *Hdac6* and *Sirt2* was not affected (Fig 2C). Furthermore, we observed that the expression of ATAT1 was reduced after knockdown of *Mettl3* (Fig 2D). Bioinformatics analysis based on the GEPIA database showed that the expression levels of METTL3 were positively correlated with ATAT1 (Fig 2E). These results indicated that METTL3 could regulate the acetylation of microtubules by targeting ATAT1. Previous studies have shown that METTL3 can bind to the gene promoters and affect their expression [34]. For this, we used a reporter system consisting of a plasmid harboring the *Atat1* promoter fragments upstream of the firefly luciferase gene. The results showed that neither wild-type METTL3 (WT METTL3) nor the catalytically dead mutant of METTL3 (mut METTL3; residues 395–398: DPPW→APPA [35]) influenced the luciferase activity (Fig 2F). The results indicated that METTL3 could not directly bind to the promoter region of *Atat1*, indicating that *Atat1* expression might be regulated by METTL3-mediated m6A modification. To assess whether METTL3-mediated m6A modification could influence ATAT1 expression, we next explored the m6A modification of *Atat1* in BMDMs. MeRIP-qPCR assays showed that the m6A modification of *Atat1* mRNA in KO BMDMs was almost unchanged (Fig 2G). RNA decay assays also showed that the half-life of *Atat1* mRNA was not affected by METTL3 (Fig 2H). These results imply that METTL3 cannot directly regulate ATAT1 expression through m6A modification.

## METTL3 depletion impairs YTHDF1-mediated translation of DNMT3A

Next, we investigated the mechanisms underlying METTL3's activity in inhibiting the production of ATAT1. We analyzed our previous m6A-seq dataset GSE146140 [36] in which *Mettl3* was knocked out specifically in myeloid cells. The top 50 m6A down-regulated genes (fold-change >4 and *p* value <0.00001) in KO BMDMs were ranked (S2A Fig). By performing Gene Ontology enrichment analysis for m6A down-regulated genes, we found that 9 of the down-regulated m6A direct targets were strongly enriched in the signaling pathway involved in transcriptional gene regulation (Fig 3A). We hypothesized that METTL3 might enhance ATAT1 expression through the enriched genes regulated by m6A modification. Further analysis demonstrated that the mRNA expression of the 9 genes enriched in transcriptional gene regulation was not affected in the KO BMDMs (Fig 3B). Bioinformatics analysis based on the GEPIA database showed that the expression levels of DNMT3A, KAT6B, and ZHX2 were strongly correlated with ATAT1 (R ≥ 0.5; *P* < 0.001) (Fig 3C). Therefore, we further analyzed the 3

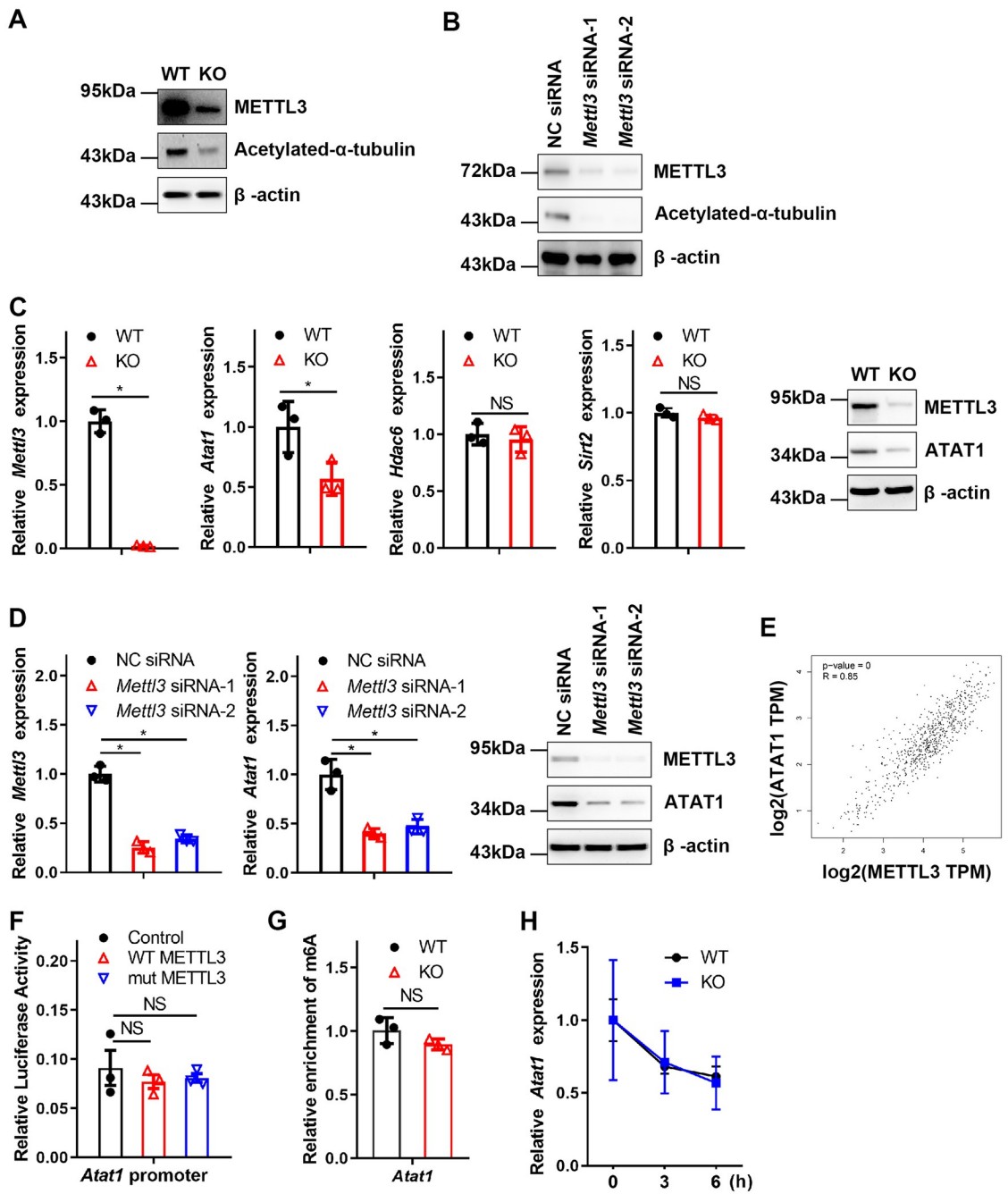

**Fig 2. METTL3 enhances ATAT1-mediated microtubule acetylation. (A)** Western blotting showing the level of α-tubulin acetylation in WT and KO BMDMs. **(B)** Immunoblotting analysis of the indicated proteins in the BMDMs transfected with NC siRNA or *Mettl3* siRNA. **(C)** Expression of *Mettl3*, *Atat1*, *Hdac6*, and *Sirt2* in WT and KO BMDMs evaluated by qRT-PCR. The expression of ATAT1 was measured by immunoblotting. **(D)** qRT-PCR and immunoblotting analysis of *Mettl3* and *Atat1* expression in BMDMs transfected with NC siRNA or *Mettl3* siRNA. **(E)** Bioinformatics analysis showed the expression levels of METTL3 and ATAT1 in normal tissue through the GEPIA database. **(F)** Cells cotransfected with a luciferase reporter construct of the *ATAT1* promoter (2 kb) and wild-type or mut METTL3. The results are shown as firefly luciferase activity normalized to Renilla luciferase activity. **(G)** MeRIP-qPCR analysis of *Atat1* expression in the WT and KO BMDMs. **(H)** *Atat1* mRNA decay assay in the WT and KO BMDMs treated with actinomycin D. All data in the figure are shown as the mean ± SD. $P < 0.05$ (*). NS means no significant difference. Underlying data can be found in S1 Data. BMDM, bone marrow-derived macrophage; KO, knockout; MeRIP-qPCR, methylated RNA immunoprecipitation-qPCR; NC, negative control; qRT-PCR, quantitative reverse transcription PCR; WT, wild type.

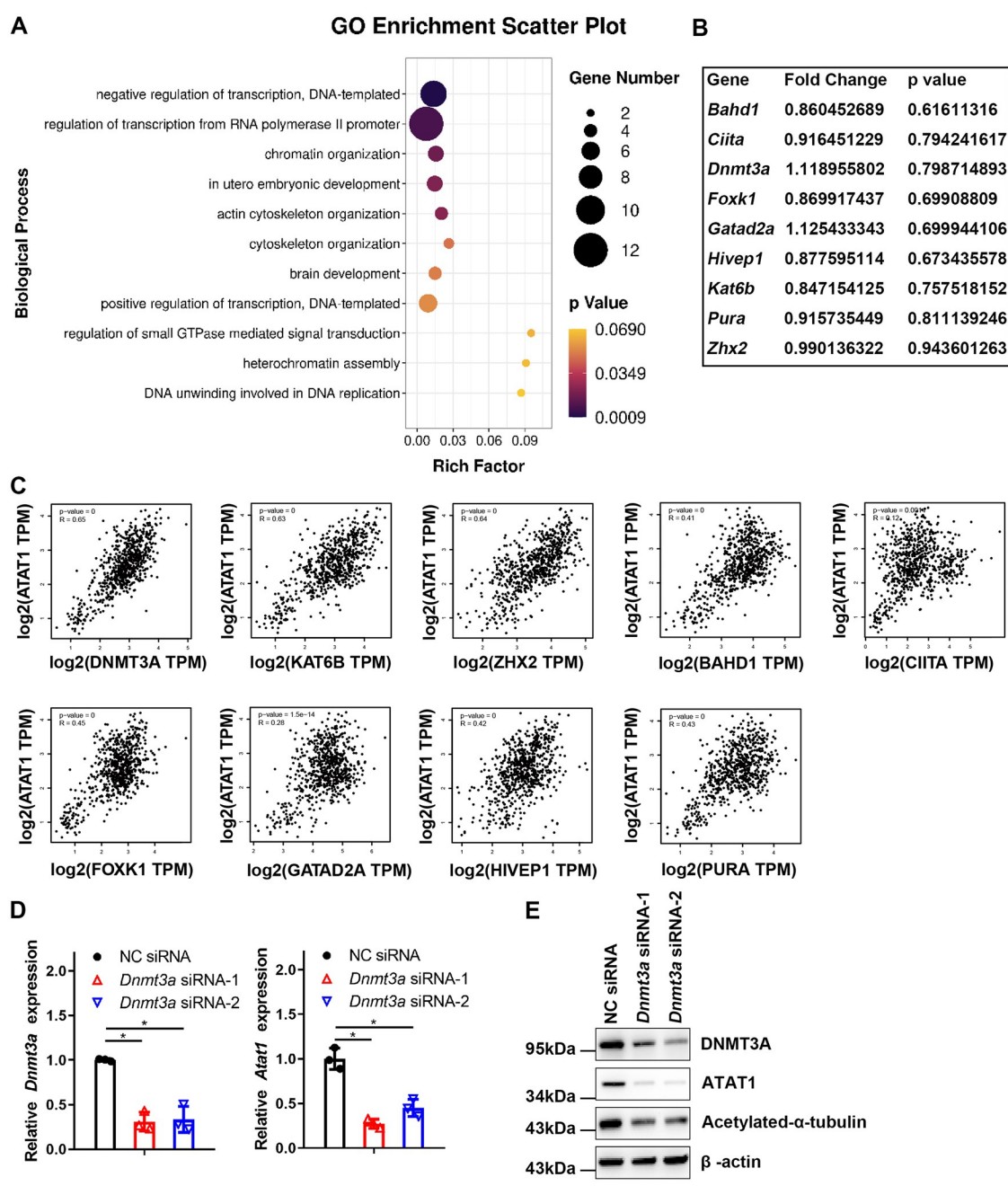

**Fig 3. DNMT3A is a target gene of METTL3 and maintains ATAT1 expression.** (**A**) GO enrichment analysis (biological process) of m6A down-regulated genes in BMDMs. (**B**) The expression of the indicated genes in KO BMDMs compared with WT BMDMs. (**C**) The association between the expression of the indicated genes and ATAT1 expression was analyzed in normal tissue through the GEPIA database. (**D**) The expression of *Atat1* was measured by qRT-PCR in WT BMDMs with *Dnmt3a* knockdown. (**E**) Immunoblotting analysis of the indicated proteins in the BMDMs transfected with NC siRNA or *Dnmt3a* siRNA. All data in the figure are shown as the mean ± SD. $P < 0.05$ (*). NS indicates no significant difference. Underlying data can be found in S1 Data. BMDM, bone marrow-derived macrophage; KO, knockout; NC, negative control; qRT-PCR, quantitative reverse transcription PCR; WT, wild type.

candidates and assessed whether *Atat1* transcription was regulated by them by RNA interference. As shown in Figs 3D and S2B, knockdown of DNMT3A, but not KAT6B or ZHX2, notably attenuated ATAT1 expression in BMDMs, which was accomplished with reduced acetylation of α-tubulin (Fig 3E). These results indicate that METTL3 may regulate ATAT1 expression through DNMT3A.

Next, we explored whether DNMT3A was a m6A-regulated target gene using m6A immunoprecipitation (m6A-IP) followed by quantitative reverse transcription PCR (qRT-PCR) and confirmed that *Dnmt3a* was a m6A-regulated target gene (Fig 4A). We next found that DNMT3A protein expression was reduced in KO BMDMs, while its mRNA level was not significantly altered (Fig 4B). Similar results were also obtained in METTL3-knockdown BMDMs (Fig 4C). In addition, mRNA decay assays showed that m6A modification did not significantly influence the decay of *Dnmt3a* mRNAs (Fig 4D). We hypothesized that down-regulation of DNMT3A protein expression in the KO BMDMs may be due to a difference in protein translation efficiency controlled by the m6A reader protein YTHDF1, which mainly promotes translation of m6A methylated transcripts [37]. Next, we extracted the RNA fractions: nontranslating fraction (<40S), translation initiation fraction (including 40S ribosomes, 60S ribosomes, and 80S monosomes) and translation active polysomes (>80S) from WT and KO BMDMs through polysome profiling. The qRT-PCR results showed that *Dnmt3a* mRNA from the *Mettl3*-deficient BMDMs in translation-active polysomes (>80S) was appreciably lower than that in the WT BMDMs (Fig 4E).

To investigate whether m6A methylation could directly regulate the expression of DNMT3A, we analyzed the conserved m6A motifs of *Dnmt3a* mRNA and then mutated the 2 potential conserved m6A motifs GGAC to GGTC (*Dnmt3a* mut1 or *Dnmt3a* mut2) respectively or (*Dnmt3a* mut1/2) simultaneously (Fig 4F). The results showed that DNMT3A protein levels, but not mRNA expression, were reduced in *Dnmt3a* mut1 and *Dnmt3a* mut2, especially in *Dnmt3a* mut1/2, which indicated that m6A motifs in both *Dnmt3a* mut1 and *Dnmt3a* mut2 were the main sites for expression regulation (Fig 4G and 4H). Furthermore, RNA-IP analysis showed that the binding between YTHDF1 and *Dnmt3a* mRNA was significantly attenuated in the KO BMDMs compared with the WT BMDMs (Fig 4I). Western blotting also showed that the expression of DNMT3A was inhibited by YTHDF1 knockdown (Fig 4J). Taken together, these data reveal that METTL3 ablation leads to the reduced expression of DNMT3A mediated by YTHDF1.

## DNMT3A transcriptionally regulates the expression of ATAT1

As an epigenetic modifier, DNMT3A usually binds directly to the gene promoter to regulate transcription [38]. Accordingly, DNMT3A was found to directly bind to both the proximal promoters and distal promoters of *Atat1* in BMDMs by chromatin-immunoprecipitation (ChIP) assay (Fig 5A). Furthermore, a previous study reported that the binding of DNMT3A to DNA might be used to maintain active chromatin states together with DNA methylation by antagonizing silencing modifications, such as trimethylation of histone H3 at Lys27 (H3K27me3) catalyzed by EZH2 in specific promoter region [39]. Our results showed that the abundance of H3K27me3 at the promoter region of *Atat1* was increased in BMDMs after knockout of METTL3 (Fig 5B). Furthermore, we assessed the ability of EZH2 to bind to the promoter region of *Atat1* and found that the binding of EZH2 to the promoter region of *Atat1* was greatly increased in the *Mettl3* knockout BMDMs (Fig 5B). To explore the mechanism underlying the regulation of ATAT1 by DNMT3A, we first analyzed the expression of EZH2 and H3K27me3 in WT and KO BMDMs. The results showed that METTL3 did not affect the abundance of EZH2 and H3K27me3 (S3A Fig). We also detected the expression of EZH2 and

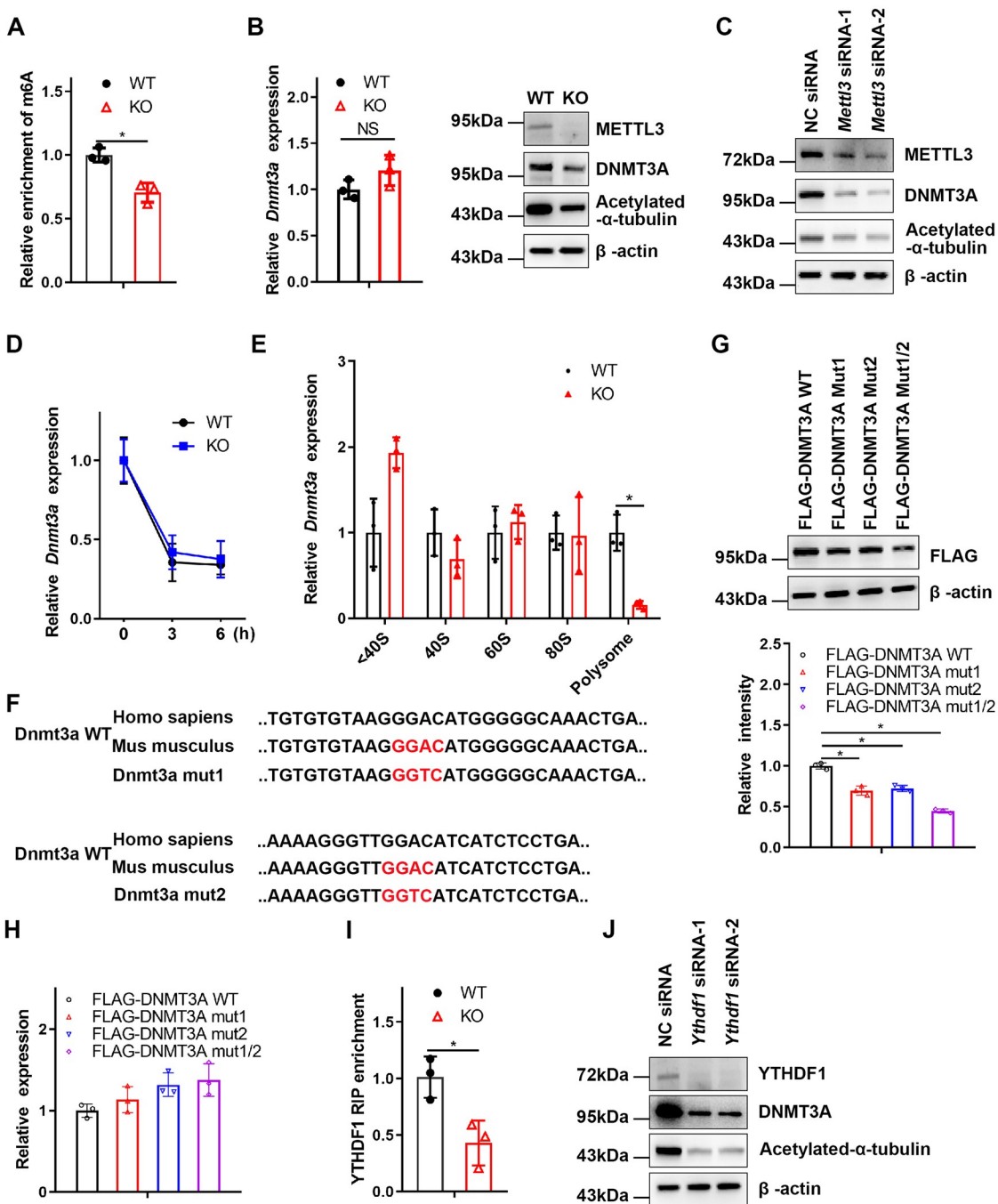

**Fig 4. METTL3 facilitates DNMT3A translation mediated by YTHDF1. (A)** MeRIP-qPCR analysis of *Dnmt3a* expression in WT and KO BMDMs. **(B)** qRT-PCR and western blotting analysis of DNMT3A levels in the WT and KO BMDMs. **(C)** Immunoblotting analysis of the indicated proteins in the BMDMs transfected with NC siRNA or *Mettl3* siRNA. **(D)** *Dnmt3a* mRNA decay assay in the WT and KO BMDMs after actinomycin D treatment. **(E)** Analysis of *Dnmt3a* mRNA in non-ribosome portion (<40S), 40S, 60S, 80S, and polysome. **(F)** Schematic representation of mutations of the conserved m6A site in *Dnmt3a* mRNA. **(G)** The expression of DNMT3A WT, DNMT3A mut1, DNMT3A mut2, and DNMT3A mut1/2 was measured by western blots. The normalized intensity of DNMT3A WT, DNMT3A mut1, DNMT3A mut2, and DNMT3A mut1/2 was quantified. **(H)** qRT-PCR analysis of the expression of DNMT3A WT, DNMT3A mut1, DNMT3A mut2, and DNMT3A mut1/2. **(I)** Binding of YTHDF1 with *Dnmt3a* mRNAs in the WT or KO BMDMs was analyzed by YTHDF1 RIP-qPCR. **(J)** Immunoblotting analysis of the indicated proteins in the BMDMs transfected with NC siRNA or *Ythdf1* siRNA. All data in the figure are shown as the mean ± SD. $P < 0.05$ (*). NS indicates no significant difference. Underlying data can be found in S1 Data. BMDM, bone marrow-derived macrophage; KO, knockout; MeRIP-qPCR, methylated RNA immunoprecipitation-qPCR; NC, negative control; qRT-PCR, quantitative reverse transcription PCR; RIP-qPCR, RNA immunoprecipitation PCR; WT, wild type.

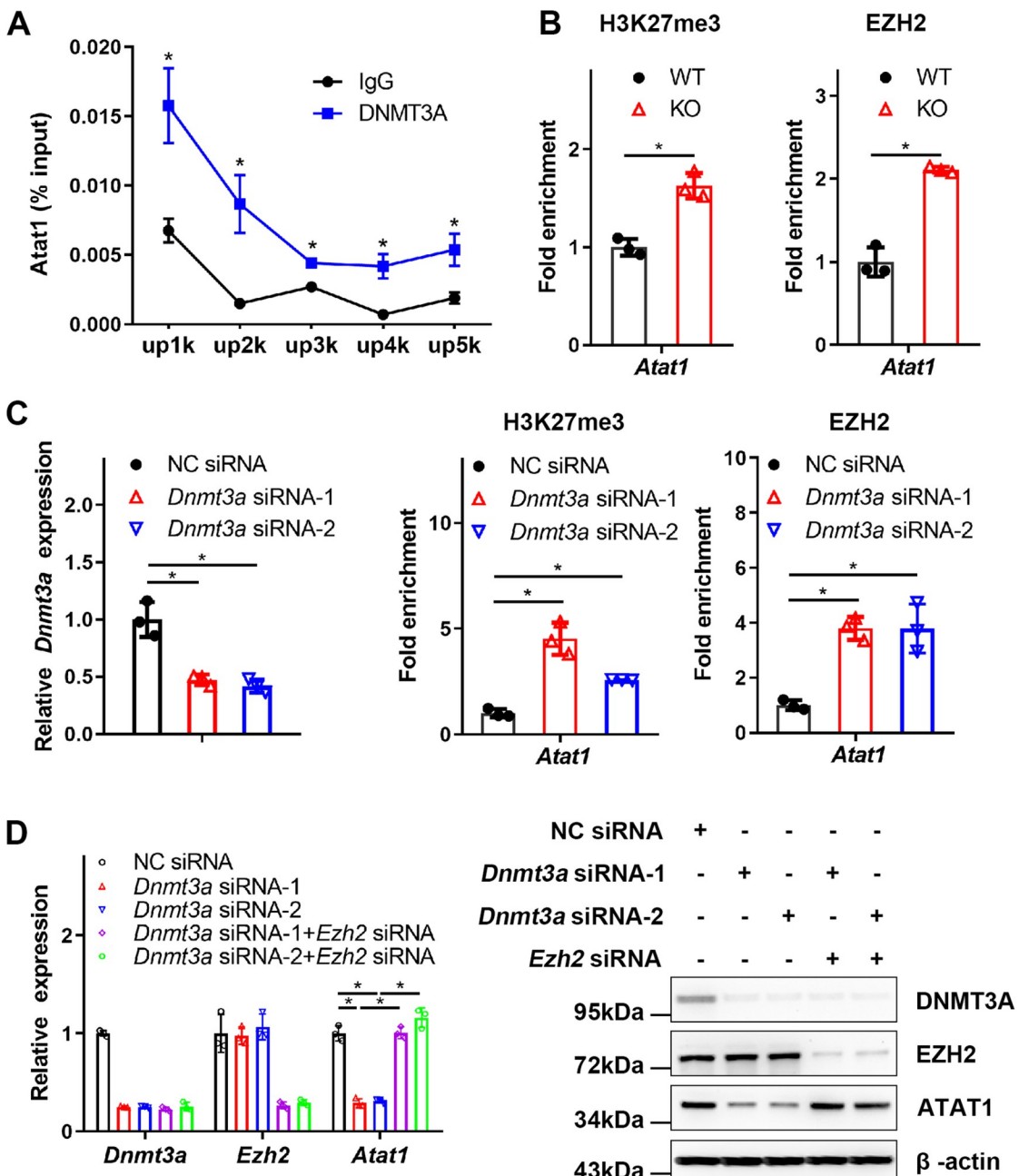

**Fig 5. DNMT3A transcriptionally regulates the expression of ATAT1. (A)** ChIP analysis of *Dnmt3a* at positions 5 kilobases (kb) (up5k), 4 kb (up4k), 3 kb (up3k), 2 kb (up2k), or 1 kb (up1k) upstream of the transcriptional start site of *Atat1* in BMDMs. **(B)** ChIP analysis of H3K27me3 or EZH2 at the *Atat1* promoter in the WT or KO BMDMs. **(C)** ChIP analysis of H3K27me3 or EZH2 at the *Atat1* promoter in the DNMT3A knockdown BMDMs. **(D)** The expression of DNMT3A, EZH2, and ATAT1 was measured by qRT-PCR and western blot in the BMDMs transfected with NC, *Dnmt3a*, or *Dnmt3a* and *Ezh2* siRNAs. All data in the figure are shown as the mean ± SD. $P < 0.05$ (*). Underlying data can be found in S1 Data. BMDM, bone marrow-derived macrophage; ChIP, chromatin immunoprecipitation; KO, knockout; NC, negative control; qRT-PCR, quantitative reverse transcription PCR; WT, wild type.

H3K27me3 in DNMT3A knockdown cells and demonstrated that DNMT3A did not significantly affect EZH2 and H3K27me3 levels (S3B Fig). These results indicated that METTL3 and DNMT3A did not regulate the expression of EZH2, which excluded that the enhanced binding of EZH2 to the promoter region of *Atat1* was caused by increased expression of EZH2. Next, we performed ChIP assays using EZH2 and H3K27me3 antibodies in DNMT3A knockdown cells to explore whether DNMT3A affects the occupation of EZH2 and H3K27me3 in the *Atat1* promoter region. The results exhibited that DNMT3A knockdown enhanced the binding capability of H3K27me3 and EZH2 in *Atat1* promoter region (Fig 5C). Previous results have demonstrated that knockdown of DNMT3A inhibited the expression of ATAT1 (Fig 3D and 3E). Furthermore, we observed that double knockdown of both DNMT3A and EZH2 inhibited the reduced expression of ATAT1 induced by DNMT3A knockdown (Fig 5D). Taken together, these results reveal that DNMT3A maintains the expression of ATAT1 by antagonizing H3K27me3 catalyzed by EZH2 in the *Atat1* promoter region.

## METTL3 depletion enhances the transmigration of brain-infiltrating macrophages involved in Aβ clearance in an Alzheimer's disease model

The above results showed that METTL3 depletion in monocyte-derived macrophages impaired α-tubulin acetylation and improved the symptoms of AD. Previous studies have reported that blood-derived monocytes adopt many functional phenotypes that contribute to progressive neurodegeneration [28]. α-Tubulin acetylation has been shown to contribute to cell migration, facilitating fibroblast, and neuronal motility [32,40,41]. These findings prompted us to investigate whether METTL3 could influence the migration of monocyte-derived macrophages into cortical brain regions. As shown in Fig 6A and 6B, the results showed that the infiltration of monocyte-derived macrophages stained with the CD45 and IBA1 markers increased in mice with Aβ-induced AD and aged mice with METTL3 depletion. Mouse brain tissue was also taken for analysis by flow cytometry. We observed that the frequency of monocyte-derived macrophage (CD45+CD11b+F4/80+) cells from the KO mice was increased significantly compared to that from the WT mice (Fig 6C). Similar results were also obtained in mice with Aβ-induced AD with *Mettl3*-deficient BM cells or monocyte transplantation (S4A and S4B Fig). In vitro transwell assays also suggested BMDMs with METTL3 depletion showed an enhanced cell migration (Fig 6D). We also found that the reduced expression of DNMT3A or ATAT1 was closely associated with a higher migration rate of BMDMs (Fig 6E and 6F). In addition, overexpression of ATAT1 in DNMT3A knockdown cells inhibited the enhanced migration (Figs 6G, S4C and S4D). Growing evidence suggests that infiltrating innate immune cells may play a key role in the clearance of Aβ in murine models of AD [42]. This finding prompted us to investigate how monocyte-derived *Mettl3*-deficient macrophages relate to the Aβ burden. We analyzed Aβ accumulation by Aβ staining in the cortical brain. The results revealed that amyloid burden was significantly decreased in the Aβ-induced AD mice and aged mice with METTL3 depletion compared to the WT mice (Fig 7A and 7B). Chimeric mice injected with *Mettl3*-deficient BM cells or monocytes also showed a decreased burden of Aβ (S5A and S5B Fig). Furthermore, we assessed the phagocytosis function of macrophages in vitro and determined that Aβ clearance was enhanced in the METTL3 deficient BMDMs in vitro (Fig 7C). Similar results were also obtained in the DNMT3A or ATAT1 knockdown BMDMs (S6 Fig). In addition, overexpression of ATAT1 inhibited the enhanced Aβ uptake induced by DNMT3A knockdown (Fig 7D). Collectively, these results imply that the m6A-DNMT3A-ATAT1 axis plays a key role in AD progression by regulating the migration and Aβ clearance of monocyte-derived macrophages (Fig 7E).

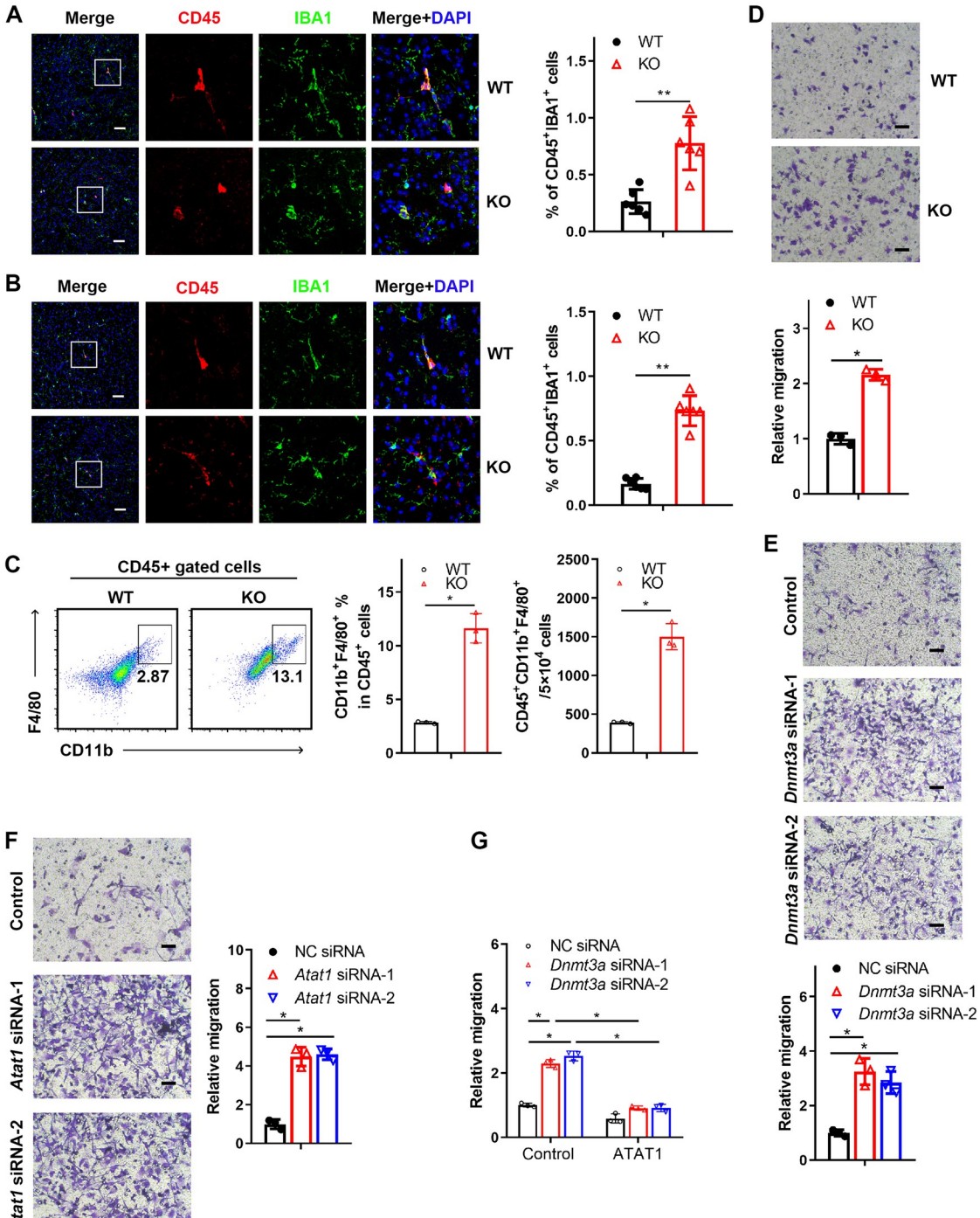

**Fig 6. METTL3 inhibits the migration and infiltration of monocyte-derived macrophage into brain.** (**A**) Colocalization of CD45 with IBA1 in the brain of Aβ-induced AD mice. Images are representative of 3 independent experiments. Scale bars: 50 μm. (**B**) Immunofluorescence showing the colocalization of endogenous CD45 with IBA1 in the brains of 24-month-old mice. The percentage of CD45+IBA1+ cells was quantified. Scale bars: 50 μm. (**C**) Flow cytometry analysis of brains from the Aβ-treated WT or KO mice. (**D**) Cell migration in transwell assays was assessed with WT and KO BMDMs. Scale bars: 30 μm. (**E**) Transwell assays were used to determine the migration of BMDMs transfected with NC siRNA or *Dnmt3a* siRNA. Scale bars: 30 μm. (**F**) Transwell assays were performed to determine the migration of the BMDMs transfected with NC siRNA or *Atat1* siRNA. Scale bars: 30 μm. (**G**) Cell migration was quantified in the ATAT1-overexpressing THP1 cells transfected with NC siRNA or *Dnmt3a* siRNA. All data in the figure are shown as the mean ± SD. $P < 0.05$ (*), $P < 0.01$ (**). Underlying data can be found in S1 Data. Aβ, amyloid beta; AD, Alzheimer's disease; BMDM, bone marrow-derived macrophage; KO, knockout; NC, negative control; WT, wild type.

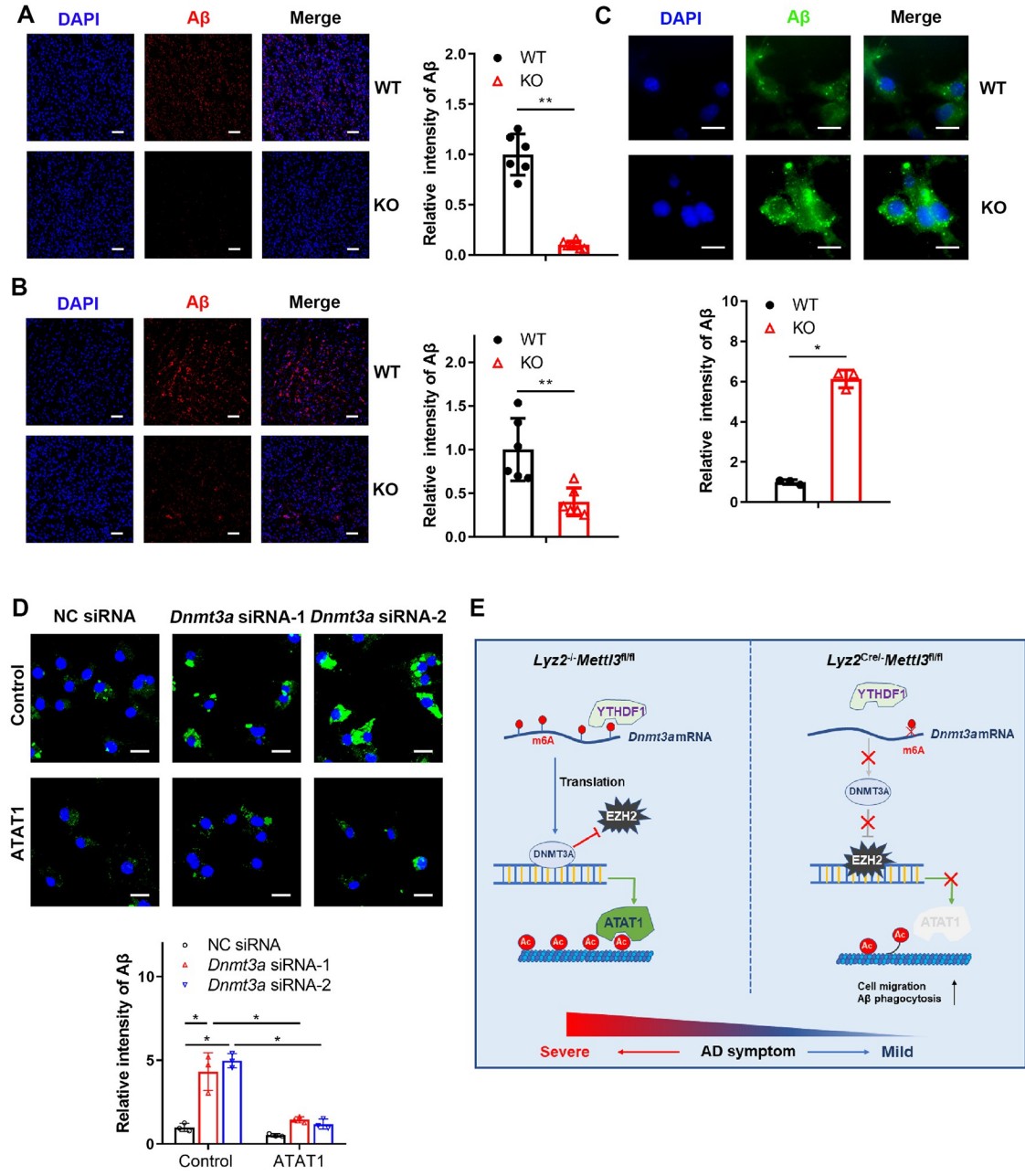

**Fig 7. METTL3 deficiency enhances Aβ uptake. (A)** Representative fluorescence micrographs of brains from Aβ-injected WT and KO mice, immunolabeled for anti-Aβ. Scale bars: 50 μm. **(B)** Representative fluorescence micrographs of brains from the KO (24 months old) and age-matched WT mice, immunolabeled with Aβ antibody. Scale bars: 50 μm. **(C)** Representative fluorescent micrographs and quantitative analysis of Aβ uptake in WT or KO BMDMs stained with Aβ antibody. Scale bars: 10 μm. **(D)** Representative fluorescence micrographs and quantitative analysis of Aβ uptake in the ATAT1-overexpressing THP1 cells transfected with NC siRNA or *Dnmt3a* siRNA. Scale bars: 10 μm. **(E)** Schematic diagram of the role of the m6A writer METTL3 in AD. All data in the figure are shown as the mean ± SD. $P < 0.05$ (*), $P < 0.01$ (**). Underlying data can be found in S1 Data. Aβ, amyloid beta; AD, Alzheimer's disease; BMDM, bone marrow-derived macrophage; KO, knockout; NC, negative control; WT, wild type.

## Discussion

In this study, we found that METTL3 deficiency increased the infiltration of monocyte-derived macrophages in an Aβ-induced AD mouse model. Further analysis demonstrated that METTL3 ablation in monocyte-derived macrophages attenuated the m6A modification in *Dnmt3a* mRNAs and subsequently impaired the translation of DNMT3A and ATAT1 expression and acetylation of tubulin, which enhanced the migration of monocyte-derived macrophages, accelerated the clearance of Aβ and alleviated AD symptoms.

A previous study showed that METTL3 is related to hippocampal memory function [21], while another study demonstrated that in brain tissues, FTO has a wide variety of physiological and pathological functions [14]. In addition, METTL14, another m6A RNA methyltransferase, is important for the transcriptional regulation of striatum function and learning epitopes [19]. Other studies have suggested that m6A RNA methylation is related to the development of the cerebellum and neural development [12,43]. A recent study showed that the dysregulation of RNA methylation is related to AD [44]. These results prompted us to explore the role of monocyte-derived macrophages regulated by m6A modification in AD. Our study will play a key role in improving our understanding of AD.

AD is characterized by gradual memory loss owing to progressive brain atrophy accompanied by amyloid plaques and neurofibrillary tangles [45–47]. Although the participation of myeloid cells in AD has been extensively researched, their contributions to disease pathology and repair are still unclear [27,48–50]. Simard and colleagues showed that bone marrow-derived mononuclear phagocytes can infiltrate into AD mouse brain and accumulate in the area of amyloid-β plaques. Transplantation of bone marrow from WT mice into AD mice was found to reduce AD pathology, supposedly relying on the phagocytic actions of bone marrow-derived cells [27]. Another study showed that AD pathology is ameliorated upon treatment of mice with macrophage colony-stimulating factor (M-CSF) after bone marrow transplantation. Fewer amyloid-β monomers were observed in the extracellular protein-enriched fractions of M-CSF-treated transgenic mice compared with vehicle controls [51]. In our study, we found that METTL3 deficiency enhanced myeloid cell infiltration into the brains of aged mice or mice with Aβ-induced AD. Further analysis showed that METTL3 deficiency improves the capability of Aβ phagocytosis. In vivo experiments also demonstrated that brain sections from KO AD mice showed a decreased burden of Aβ.

ATAT1 catalyzes the acetylation of α-tubulin at lysine 40 in various organisms ranging from tetrahymena to humans [52]. A very preliminary observation shows that the lateral ventricles are large in *Atat1*$^{-/-}$ mice [53]. Our results demonstrated that decreased α-tubulin acetylation was closely associated with reduced expression of METTL3 in BMDMs. α-Tubulin acetylation has been shown to contribute to cell migration facilitating fibroblast and neuronal motility [32,40,41]. We found that METTL3 depletion enhanced the infiltration of BMDMs into the AD mouse brain. In vitro experiments also showed that METTL3 depletion enhanced the migration of BMDMs. Previous studies have shown that m6A modification is a versatile regulator of mRNA stability, splicing, localization, and translation rate [5,7]. Mechanistically, we found that METTL3 depletion impaired YTHDF1-mediated translation of DNMT3A. DNMT3A usually binds directly to the gene promoter to regulate transcription [38]. Furthermore, a previous study demonstrated that the binding of DNMT3A to DNA contributes to maintaining active chromatin states together with DNA methylation by inhibiting silencing modifications, such as EZH2-mediated trimethylation of histone H3 at Lys27 (H3K27me3) in specific promoter regions [39]. Consistently, our results showed that DNMT3A could bind to the promoter region of *Atat1* and maintain its expression by antagonizing H3K27me3 modification.

In summary, we discovered that m6A-DNMT3A-ATAT1 can regulate the acetylation of α-tubulin in BMDMs. METTL3 deficiency enhances the infiltration of monocyte-derived macrophages into the AD brain and Aβ clearance, subsequently improving the cognitive decline. Our results suggest that m6A modifications are potential targets for the treatment of AD. However, the upstream regulation of METTL3 in AD remains to be explored in the future. At present, some clinical trials targeting Aβ are investigated for the treatment of early AD [54,55]. Our results may supply a new choice for AD treatment strategies.

## Methods

### Ethics statement

All animal care and use protocols were performed in accordance with the Regulations for the Administration of Affairs Concerning Experimental Animals approved by the State Council of People's Republic of China. The animal experiments were approved by the Institutional Animal Care and Use Committee of Xinxiang Medical University (Approval Number: XXLL-2018B005) and the Animal Experiment Administration Committee of Fourth Military Medical University (approval number: IACUC-20221210).

### Mice

$Mettl3^{fl/fl}Lyz2^{-/-}$ (WT) and $Mettl3^{fl/fl}Lyz2^{cre/-}$ (KO) mice were generated as previously described [36]. All experimental mice were maintained under specific pathogen-free conditions, fed standard laboratory chow, and kept on 12 h light to 12 h dark cycles. The temperature and humidity were set at 22 ± 1°C and 55% ± 5%, respectively. Cohoused Cre-negative littermate mice were used as control animals in all experiments.

### Cell lines

HEK293T cells were maintained in Dulbecco's Modified Eagle's Medium (DMEM) containing 10% fetal bovine serum (Gibco), 100 U/ml penicillin, and 100 μg/ml streptomycin (Gibco). The cells were cultured in a 5% $CO_2$ humidified incubator at 37°C. The THP1 cells were cultured at 37°C in an atmosphere of 5% $CO_2$ and maintained in RPMI 1640 medium (Gibco) with 10% fetal bovine serum (Gibco) with 100 U/ml penicillin and 100 μg/ml streptomycin (Gibco).

### Plasmid construction

The ATAT1 promoter was amplified and cloned into the PGL-3.0 vector. FLAG-DNMT3A was constructed by cloning full-length mouse DNMT3A cDNA into the p3×FLAG-CMV-10 vector. FLAG-DNMT3A mut1, FLAG-DNMT3A mut2, or FLAG-DNMT3A mut1/2 were generated by mutating the 2 potential conserved m6A motifs GGAC to GGTC separately or simultaneously. HA-ATAT1 was generated by cloning full-length ATAT1 cDNA into the pCDH vector.

### Bone marrow-derived macrophages

Bone marrow was isolated from femurs and tibia of mice by flushing. The cells were cultured in DMEM with 10% fetal bovine serum, streptomycin (100 μg/ml), penicillin (100 U/ml), and M-CSF (40 ng/ml). Cells were cultured for 7 days to generate BMDMs. Cultured BMDMs were used for qRT-PCR analysis, western blotting, and other experiments.

## Intracerebroventricular injection

The experiments were performed as previously described [56]. Aβ1–42 (Anaspec, San Jose, California) was reconstituted in sterile saline solution as a stock solution at a concentration of 100 μm, followed by incubation at 37˚C for 4 days. Animals were anesthetized with isoflurane, and the site of injection was stereotaxically reached. The injection point was located 1.8 ± 0.1 mm lateral to the sagittal suture, −1.0 ± 0.06 mm posterior to the bregma, and 2.4 mm in depth. Five microliters of solution containing either 0.9% NaCl or recombinant Aβ1–42 was injected over a period of 3 min.

## Assay for Aβ phagocytosis

BMDMs were seeded at 10,000 cells per well in glass slides. Two days later, the cells were incubated for 12 h with 1 μm Aβ. Thereafter, the cells were rinsed several times with PBS and then fixed with 4% formaldehyde for 15 min at room temperature. The cells were then rinsed 3 times with PBS and probed overnight at room temperature with antibody against Aβ in PBS containing 0.1% Triton X-100, followed by Alexa Fluor-labeled secondary antibodies for 0.5 h at room temperature. Fluorescence images were acquired on a laser confocal microscope (Nikon).

## Transfection

For siRNA transfection, cells were seeded 1 day before transfection. Transfection was performed when the cell confluency was approximately 70% to 80% using Lipofectamine RNAi-MAX (Thermo Fisher Scientific) following the manufacturer's instructions. The siRNA duplex oligonucleotides used are listed in S1 Table. Lipofectamine 2000 (Thermo Fisher Scientific) was used for plasmid transfections following the manufacturer's instructions.

## Quantitative real-time PCR (qRT-PCR)

Total RNA was isolated from cells using TRIzol (Thermo Fisher Scientific). For qRT-PCR analysis of mRNAs, cDNA was generated using a PrimeScript One Step RT-PCR Kit (TaKaRa). The qRT-PCR experiment was conducted using a SYBR Premix E×Taq Kit (TaKaRa). The primers used for qRT-PCR are listed in S2 Table.

## Immunoblotting

Cells were lysed in RIPA buffer (1% Triton X-100, 20 mM Na2PO4, 150 mM NaCl (pH 7.4)) containing PMSF and Phosphatase Inhibitor Cocktail (Roche). Subsequently, a BCA assay (Thermo Fisher Scientific) was used to determine the protein concentrations. Proteins were separated by SDS-PAGE gels and transferred onto nitrocellulose membranes (Millipore). The membranes were placed in TBST (10 mM Tris-HCl (pH 7.4), 150 mM NaCl, and 0.1% Tween-20) containing 5% nonfat milk for 1 h at room temperature. Primary antibodies were diluted in TBST containing 5% BSA and used at the indicated concentrations: rabbit anti-acetyl-α-tubulin (1:1,000, 5335, Cell Signaling Technology), rabbit anti-EZH2 (1:1,000, 5246, Cell Signaling Technology), rabbit anti-H3K27me3 (1:1,000, 9733, Cell Signaling Technology), rabbit anti-DNMT3A (1:1,000, ab2850, Abcam), rabbit anti-YTHDF1 (1:1,000, 17479-1-AP, 18810, Proteintech), rabbit anti-METTL3 (1:1,000, ab18810, Abcam), and mouse anti-β-actin (1:5,000, Sigma). The membranes were incubated with primary antibodies overnight at 4˚C, washed with TBST 4 times and incubated with HRP-conjugated anti-mouse IgG (1:10,000, 7076, Cell Signaling Technology) or anti-rabbit IgG (1:10,000, 7074, Cell Signaling Technology) diluted in TBST at room temperature for 1 h. After 4 final washes with TBST, the

membranes were developed by using ECL and visualized using Tanon 5500 or Amersham Imager 680.

## Chromatin immunoprecipitation (ChIP) assay

ChIP analysis of DNMT3A/EZH/H3K27me3 was performed using the SimpleChIP Enzymatic Chromatin IP Kit (Cell Signaling Technology) following the manufacturer's instructions. Primer sequences for ChIP analysis are listed in S2 Table.

## Luciferase assay

The luciferase assay was performed as previously described [57]. Briefly, HEK293T cells were seeded in 48-well plates at a density of 5,000 cells per well. After 24 h, the cells were transfected with 5 ng of pRL-TK Renilla luciferase reporter, 100 ng required plasmids, and 100 ng of luciferase reporter with the target gene promoter. After 48 h, luciferase activity was determined with the dual luciferase reporter assay system (Promega).

## Immunofluorescence staining

For confocal microscopy, cells were fixed in 4% paraformaldehyde after treatments, permeabilized with 0.3% Triton X-100, blocked with 5% bovine serum albumin, and then incubated with primary antibodies overnight at 4°C. Next, the cells were incubated with fluorescent dye-labeled secondary antibodies: Donkey Anti-Rat IgG Alexa Fluor 594 (1:1,000, A-21209, Thermo Fisher Scientific), Donkey Anti-Rabbit IgG Alexa Fluor 555 (1:1,000, A-31572, Life Technologies), Goat Anti-Rabbit IgG Alexa Fluor 488 (1:1,000, A-11008, Life Technologies), Goat Anti-Mouse IgG Alexa Fluor 488 (1:1,000, A-11001, Thermo Fisher Scientific), and Goat Anti-Mouse IgG Alexa Fluor 555(1:1,000, A-21422, Thermo Fisher Scientific). Nuclei were stained with DAPI. Confocal fluorescence images were captured using a Nikon confocal microscope.

## meRIP-qPCR

Total RNA was isolated from cells using TRIzol (Thermo Fisher Scientific). For meRIP-qPCR, poly (A)+ mRNA was extracted by using the Dynabeads mRNA Direct Purification Kit (61012, Thermo Fisher Scientific). m6A-containing mRNA enrichment was carried out using the Magna MeRIP m6A kit (17–10499, Millipore) from 2 μg of purified poly (A)+ mRNA, and the enriched RNA was purified according to the manufacturer's protocol. The final product was used for qRT-PCR to determine the enrichment of m6A on gene transcripts. The primers used for testing *Dnmt3a* and *Atat1* mRNA are listed in S2 Table.

## RNA immunoprecipitation-qPCR (RIP-qPCR)

This procedure was performed according to a previously published report [36]. BMDMs were washed twice with PBS and lysed in lysis buffer (150 mM KCl, 10 mM HEPES (pH 7.6), 0.5% NP-40, 2 mM EDTA, 0.5 mM dithiothreitol (DTT), protease inhibitor cocktail, 400 U/ml RNase inhibitor). The cell lysates were centrifuged to obtain the supernatant. A 50-μl aliquot of cell lysate was saved to serve as input, and the remaining lysate was incubated with 20 μl of protein A beads, previously bound to IgG antibody or anti-YTHDF1 antibody (Proteintech) for 4 h at 4°C. The beads were then washed 4 times with washing buffer (50 mM Tris, 200 mM NaCl, 2 mM EDTA, 0.05% NP40, 0.5 mM DTT, RNase inhibitor). RNA was eluted from the beads with 50 μl of elution buffer (5 mM Tris-HCL (pH 7.5), 1 mM EDTA, 0.05% SDS, 20 mg/ml Proteinase K) for 2 h at 50°C, and purified with Qiagen RNeasy columns. RNA was eluted

in 100 μl of RNase-free water and were reverse transcribed into cDNA using a PrimeScript qRT-PCR kit (TaKaRa) according to manufacturer's instructions. The fold enrichment was measured by qRT-PCR. The primers used for testing *Dnmt3a* mRNA are listed in S2 Table.

## mRNA stability analysis

For determination of mRNA stability, BMDMs were treated with actinomycin D (Sigma) at a final concentration of 5 μg/ml for 0, 3, or 6 h. The cells were then collected, and total RNA was extracted for reverse transcription using a PrimeScript qRT-PCR kit (TaKaRa). The mRNA transcript levels of interest were determined by qRT-PCR.

## Polysome profiling analysis

Polysome profiling was performed as previously published [58]. BMDMs were first treated with cycloheximide (0.1 mg/ml) for 3 min at room temperature to arrest and stabilize polysomes, washed with PBS, and then lysed with 1 ml of cold polysome lysis buffer (0.3 M NaCl; 15 mM MgCl2.6H2O; 15 mM Tris-HCl (pH 7.4)) containing 10 μl of Triton X-100 (1%), 1 μl of 100 mg/ml cycloheximide in DMSO and RNasin. The cell lysates were centrifuged at 13,000 rpm for 15 min at 4°C. The supernatants were fractionated by 10% to 50% sucrose gradient centrifugation (35,000 rpm for 1.5 h) in a Beckman ultracentrifuge. Each fraction of the density gradient was collected, and the absorbance was detected at 260 nm. Ribosomal RNA content measured at 260 nm was plotted to obtain the polysome profile of each sample. RNA from each fraction was isolated and reverse transcribed using a PrimeScript qRT-PCR kit (TaKaRa). The obtained cDNAs were amplified using real-time PCR analysis for polysome abundance on *Dnmt3a* mRNAs.

## Flow cytometry

For brain tissue analysis, animals were sublethally anesthetized by intraperitoneal injection of pentobarbital. Mice were then perfused transcardially with sterile cold saline solution until the blood was completely washed out. Brains were quickly removed and placed in cold medium and then dissociated using a Neural Tissue Dissociation kit (Miltenyi Biotec) according to the instructions of the manufacturer. Single-cell suspensions were analyzed using the following antibodies: CD45.2-PerCP (Biolegend), CD11b-PE-cy7 (BD Bioscience), F4/80-PE (BD Bioscience), Ly6C-FITC (BD Bioscience), and CD115-PerCP (Biolegend) and subjected to flow cytometry analysis.

## Bone marrow chimeras

BM chimeras were prepared as previously described [59]. In brief, mice were exposed to irradiation (9 Gy). The bone marrow cells were aseptically collected from WT or KO mice by flushing femurs and tibia with Dulbecco's PBS with 2% fetal bovine serum. The samples were filtered through a 40-μm nylon mesh and centrifuged. Irradiated animals were injected via a tail vein with $1 \times 10^7$ bone marrow cells, housed in autoclaved cages, and treated with antibiotics 2 weeks after irradiation. After 5 weeks, the chimeric mice were subjected to the AD model.

## Monocytes transplantation

Monocyte transplantation was performed as previously described [60]. In brief, bone marrow cells were collected from WT or KO (CD45.2) donor mice by flushing the femurs and tibia. Cells were passed through a 40-μm pore nylon mesh and rinsed with DPBS containing 2% FBS. CD115+ cells were isolated using a magnetic cell separation system and biotinylated anti-

CD115 antibody combined with streptavidin-magnetic beads (Miltenyi Biotec). Then, the isolated monocytes were injected via the tail vein into mice.

## Morris water maze (MWM)

For evaluation of spatial learning and memory after Aβ injections, the MWM test was performed as previously described [61]. Briefly, a circular pool (diameter, 120 cm; height, 30 cm) located in a large room with various distal visual cues was filled with water. A circular platform (diameter, 10 cm) was located at a fixed position approximately 1 cm below the water surface. The swimming activity of each mouse was recorded. In the training phase, the mice were given 4 trials to locate a visible platform for 4 consecutive days starting from different positions. In each of the 4 trials, the mouse was allowed 60 s to find the platform. If the mouse could not find the platform within 60 s, it was guided to the platform and allowed to stay there for 20 s. On Day 5, the platform was removed from the pool in the probe trial. The latency to the first platform area crossing, the number of crossings, and the time spent in the platform quadrant were recorded in 60 s.

## Open-field test

The mouse activity was recorded to study locomotor and exploratory behavior as previously described [62]. Mice were allowed to explore in a gray square box for 5 min. The velocity and time spent in the center area were recorded.

## Y maze test (YMT)

Working memory and exploratory activity were measured using a Y-maze apparatus as previously described [63]. The arms of the apparatus were 30 cm long, 8 cm wide, and 15 cm high. Each mouse was placed at the end of 1 arm. The number of alterations was recorded for 5 min. Working memory was calculated as the number of correct alterations/number of total arm entries. Correct alternation means entry into all 3 arms on consecutive choices (i.e., ABC, BCA, or CAB, but not CAC, BAB, or ABA).

## Statistical analysis

Most experiments were repeated at least twice. Statistical significance was determined by the Mann–Whitney test, and differences were considered statistically significant when $P < 0.05$ (\*), $P < 0.01$ (\*\*), or $P < 0.001$ (\*\*\*).

## Supporting information

**S1 Fig. METTL3 deficiency attenuates the cognition impair in a mouse model of Alzheimer's disease.** (**A**) Flow cytometry of the surface CD11b and F4/80 expression of primary cultured bone marrow-derived macrophages. (**B**) Aβ-injected WT and KO mice were assessed in the open field test. Data are presented as a mean ± SEM. (**C**) The purity of isolated monocytes was checked by flow cytometry. (**D**) Flow cytometry analysis of brains from Aβ-induced WT or KO mice injected with CCR2 antagonist PF-4136309 (2 mg/kg). $P < 0.05$ (\*). NS means no significant difference. Underlying data can be found in S1 Data. Aβ, amyloid beta; KO, knockout; WT, wild type.
(TIF)

**S2 Fig. DNMT3A is a target gene of METTL3 and regulates ATAT1 expression.** (**A**) The ranking of top 50 m6A down-regulated genes in KO BMDMs. (**B**) The expression of *Atat1* was measured by qRT-PCR in BMDMs with *Kat6b* or *Zhx2* knockdown. Data in the figure are

shown as mean ± SD. $P < 0.05$ (*). NS means no significant difference. Underlying data can be found in S1 Data. KO, knockout; qRT-PCR, quantitative reverse transcription PCR.
(TIF)

**S3 Fig. The expression of EZH2 and H3K27me3 is not regulated by METTL3 or DNMT3A.** **(A)** Immunoblotting analysis of the indicated proteins in WT and KO BMDMs. (**B**) Immunoblotting analysis of the indicated proteins in BMDMs transfected with NC siRNA or *Dnmt3a* siRNA. Underlying data can be found in S1 Data. KO, knockout; NC, negative control; WT, wild type.
(TIF)

**S4 Fig. METTL3-deficiency enhances the migration of macrophage. (A)** Flow cytometry analysis of brains from Aβ-induced mice transferred with WT and KO BM. (**B**) Flow cytometry analysis of brains from Aβ-induced mice (CD45.1) transferred with WT or KO monocytes (CD45.2). **(C)** Transwell assays were performed to determine the migration of ATAT1-overexpressing BMDMs transfected with NC siRNA or *Dnmt3a* siRNA. Scale bars: 30 μm. **(D)** Immunoblotting analysis of the indicated proteins in ATAT1-overexpressing BMDMs transfected with NC siRNA or *Dnmt3a* siRNA. Data in the figure are shown as mean ± SD. $P < 0.05$ (*), $P < 0.01$ (**). NS means no significant difference. Underlying data can be found in S1 Data. Aβ, amyloid beta; KO, knockout; NC, negative control; WT, wild type.
(TIF)

**S5 Fig. METTL3-deficient macrophage enhances the phagocytosis of Aβ. (A)** Representative fluorescence micrographs and quantitative analysis of brains from Aβ-induced mice transferred with WT and KO BM. (**B**) Representative fluorescence micrographs and quantitative analysis of brain from Aβ-induced mice transferred with WT and KO monocytes. Scale bars: 50 μm. Data in the figure are shown as mean ± SD. $P < 0.01$ (**). Underlying data can be found in S1 Data. Aβ, amyloid beta; KO, knockout; WT, wild type.
(TIF)

**S6 Fig. Knockdown of DNMT3A or ATAT1 enhances the phagocytosis of Aβ. (A)** Representative fluorescent micrographs and quantitative analysis of Aβ uptake for 24 h in BMDMs transfected with NC siRNA or *Dnmt3a* siRNA and stained with Aβ antibody. Scale bars: 10 μm. **(B)** Representative fluorescent micrographs and quantitative analysis of Aβ uptake for 24 h in BMDMs transfected with NC siRNA or *Atat1* siRNA and stained with Aβ antibody. Scale bars: 10 μm. Data in the figure are shown as mean ± SD. $P < 0.05$ (*). Underlying data can be found in S1 Data. Aβ, amyloid beta; NC, negative control.
(TIF)

**S1 Table. siRNA sequences used in this manuscript.**
(DOCX)

**S2 Table. qRT-PCR primers used in this manuscript.**
(DOCX)

**S1 Raw images. Original scan images for Figs 2A–2D, 3E, 4B, 4C, 4G, 4J, 5D, S3A, S3B, and S4D.**
(PDF)

**S1 Data. Underlying data for Figs 1–7 and S1, S2, S4, S5, and S6.**
(XLSX)

## Acknowledgments

We thank Prof. Wenjing Luo and Zipeng Cao from Fourth Military Medical University for support of the animal behavioral experiments. We are grateful to Mrs. Fangli Gao from Fourth Military Medical University for carefully breeding and reproducing the mice.

## Author Contributions

**Conceptualization:** Huilong Yin, Angang Yang, Rui Zhang.

**Funding acquisition:** Huilong Yin, Angang Yang, Rui Zhang.

**Investigation:** Huilong Yin, Zhuan Ju, Minhua Zheng, Xiang Zhang, Wenjie Zuo, Yidi Wang, Yingran Peng, Jiadi Li.

**Methodology:** Huilong Yin, Minhua Zheng, Xiaochen Ding, Xiaofang Zhang.

**Supervision:** Angang Yang, Rui Zhang.

**Validation:** Huilong Yin, Zhuan Ju.

**Visualization:** Huilong Yin, Zhuan Ju.

**Writing – original draft:** Huilong Yin, Angang Yang, Rui Zhang.

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
