## [Editor Report · Decision Letter 0]

5 Aug 2022

Dear Dr Zhang, 

Thank you for submitting your manuscript entitled "N6-methyladenosine in monocyte-derived macrophage is involved in a mouse model of Alzheimer's disease" for consideration as a Research Article by PLOS Biology.

Your manuscript has now been evaluated by the PLOS Biology editorial staff, as well as by an academic editor with relevant expertise, and I am writing to let you know that we would like to send your submission out for external peer review.

Once your full submission is complete, your paper will undergo a series of checks in preparation for peer review. After your manuscript has passed the checks it will be sent out for review. To provide the metadata for your submission, please Login to Editorial Manager (https://www.editorialmanager.com/pbiology) within two working days, i.e. by Aug 09 2022 11:59PM.

Kind regards,

Lucas

Lucas Smith, Ph.D.

Associate Editor

PLOS Biology

lsmith@plos.org

---

## [Decision Letter · Decision Letter 1]

12 Sep 2022

Dear Dr Zhang,

Thank you for your patience while your manuscript "N6-methyladenosine in monocyte-derived macrophage is involved in a mouse model of Alzheimer's disease" was peer-reviewed at PLOS Biology. It has now been evaluated by the PLOS Biology editors, an Academic Editor with relevant expertise, and by several independent reviewers. In light of the reviews, which you will find at the end of this email, we would like to invite you to revise the work to thoroughly address the reviewers' reports.

As you will see from their comments, the reviewers appreciate the importance of the topic and Reviewer 2 has suggested we accept the manuscript, as is. Despite this, both Reviewers 1 and 3 have raised a number of important and overlapping concerns with the study, which call in to question the strengths of the conclusions. Reviewers 1 and 3 also emphasize that the manuscript would benefit from a careful edit for language, clarity, and to provide additional context and justifications for this work. We feel that these concerns would need to be thoroughly addressed before we can consider you manuscript further for publication, and we would like to emphasize the need to tease apart the effects of METTL3 KO on microglia vs macrophages.

Given the extent of revision needed, we cannot make a decision about publication until we have seen the revised manuscript and your response to the reviewers' comments. Your revised manuscript is likely to be sent for further evaluation by all or a subset of the reviewers.

**IMPORTANT - SUBMITTING YOUR REVISION**

*Re-submission Checklist*

*Published Peer Review*

*PLOS Data Policy*

*Blot and Gel Data Policy*

Sincerely,

Lucas

Lucas Smith, Ph.D.

Associate Editor

PLOS Biology

lsmith@plos.org

REVIEWS:

Reviewer #1, Hua-Bing Li (note, this reviewer has signed the review): 

In this manuscript, the authors investigated the role of m6A RNA methylation in monocyte-derived macrophage involved in the progression of AD. They reported that METTL3 ablation in myeloid cells improved cognition function in AD mouse model. Mechanistically, the authors demonstrated that METTL3 promoted the translation of DNMT3A mediated by YTHDF1. DNMT3A bounded at the promoter region of Atat1 and maintained its expression. Thus, loss of METTL3 resulted in decreased translation of DNMT3A and subsequently downregulated the expression of Atat1. The decreased expression of Atat1 led to reduced �-tubulin acetylation and enhanced monocyte-derived macrophage migration, A� clearance and relief of AD symptoms. 

There are several main concerns with this manuscript: 

Major issues:

1. Introduction, page 3, line 65, the authors termed "blood-derived myeloid cells recruited by CNS damage" as " blood-derived microglia ". Recent studies show that microglia are tissue resident and originate in the yolk sac but not in the bone marrow from hematopoietic stem cells. Microglia and macrophages represent two related cell types involved in the brain pathology. The authors should replace the phrase " blood-derived microglia ". 

2. The author utilized using a Mettl3 conditional mouse line in combination with Lyz2-Cre driver to elucidate the role of m6A mRNA methylation in monocyte-derived macrophage involved in the progression of AD ( Results, page 4, line 87 ). In this system, METTL3 is also deleted in microglia. Microglial phagocytosis ability is proved to be associated with A� deposition and affect AD progression. The authors should also analyze and exclude the function of METTL3 in microglia in their system.

3. Figure1: The authors assessed the cognition function and locomotor behavior by Morris water maze test and open-field test respectively. They found that METTL3 ablation in myeloid cells improved the cognition function but not locomotor behavior. Additional behavioral experiments such as Y maze test need to be done to comprehensively evaluate the role of METTL3 in monocyte-derived macrophage involved in AD progression.

4. Fig. 2C, 2D and Fig. 3D: The expression of ATAT1 needs to be detected in both mRNA level and protein level.

5. Fig.3: The authors picked nine of the downregulated m6A direct targets from their previous m6A-seq data set. The authors need to exhibit the ranking of these genes among all the differential targets in their m6A-seq data set, and to explain why they focus on these nine genes.

6. Fig 5: The author concluded that DNMT3A maintained the expression of ATAT1 by antagonizing H3K27me3 catalyzed by EZH2 in Atat1 promoter region. Whereas, the results in Fig5 cannot support their conclusion. They showed METTL3 deletion increased the abundance of H3K27me3 and EZH2 binding at the promoter region of Atat1. Is is possible that METTL3 directly regulates the expression of EZH2 or the capability of EZH2 binding to the promoter region of Atat1. The authors should provide direct evidence that DNMT3A affects H3K27me3 catalyzed by EZH2 in Atat1 promoter region to promote ATAT1 expression. 

7. Fig 6A: The authors should analyze both percentage and absolute numbers of macrophages in the CNS through flow cytometry. 

8. Fig 6: The authors demonstrated that METTL3 depletion enhanced transmigration of brain-infiltrating macrophage. The authors should assess the impact of enhanced migration of macrophages on the progress of AD by using CCR2 blocking antibody in their system.

Minor issues:

1. Statistical methods: The pairwise comparisons should be analyzed for significance with nonparametric (Mann-Whitney) tests rather than the t-test, because of the small numbers of repeatsand the non-normally distributed data points where they are shown. 

2. I would suggest the authors to have the manuscript language edited by a native English speaker/professional.

Reviewer #2: 

The paper satisfies all the requirements for publication and can be accepted as it stands

Reviewer #3: 

Yin et al. documented a long chain of observations after cell-type-specific Mettl3 deletion with the Lyz2-Cre driver. Specifically,

1. Mettl3 deletion with the Lyz2-Cre driver  

2. Reduced m6A modification of Dnmt3a mRNA in monocyte-derived macrophages 

3. Reduced YTHDF1 binding to Dnmt3a mRNA 

4. Reduced translation of Dnmt3a mRNA to DNMT3A protein 

5. Reduced antagonization of EZH2 by DNMT3A, increased trimethylation of histone H3 in Atat1 promoter region 

6. Reduced transcription of Atat1 

7. Reduced acetylation of alpha-tubulin 

8. Enhanced infiltration of monocyte-derived macrophages into the brain 

9. Enhanced A-beta clearance 

10. Improved cognition in A-beta based AD model

The strength of the study is its potential significance in treating AD. Another strength is in proposing very specific hypotheses and in conducting studies on many of the downstream pathways. Unfortunately, for the same reason, it's an enormous burden to prove that each step in this very long chain of events is direct, specific, and causal. I will list these concerns below. 

1) There are studies showing that Lyz2 is also expressed in microglia in addition to myeloid cells. It's also expressed in other non-macrophage blood cells. How are the authors sure that the effects are mediated solely by monocyte-derived macrophages? 

2) For AD-related cognitive impairments, only water maze was used. In the water maze test, only probe trial data was used. There's no data on learning. This is very limited, especially because the whole story relies on the claim that the treatment reduces cognitive impairments.

3) Fig 3B shows very small and non-significant changes in Dnmt3a (p=0.80)

4) In Fig 4G, reduced DNMT3A level is not clear. There's no statistics either. If the argument is reduced translation, is the mRNA level normal to begin with in this experiment? 

5) DNMT3A can affect the transcription of many genes. How do we know that all the downstream effects are mediated by Atat1?

6) Acetylation of alpha-tubulin has many functional consequences. How do we know that infiltration of monocyte-derived macrophages into the brain is responsible for the A-beta and behavioral phenotype?

7) There is no direct evidence of enhanced infiltration of monocyte-derived macrophages into the brain. 

8) In Fig 6B, the Y axis is not explained. What does this "%" mean?

9) What is the normal "good" function of m6A in monocyte-derived macrophages? How do they become maladaptive in AD? 

10) M6A is known to be essential in cell differentiation, which has been shown in many cell types. Does Mettl3 deletion affect macrophage differentiation? 

Overall, the manuscript is not clearly written. In one short paragraph, "blood-derived microglia", "blood-derived monocytes", and "monocyte-derived macrophages" were all used. And in most other places, "bone marrow derived macrophages" were used. Do these four mean the same thing? Also, is "blood-derived microglia" even correct? 

Overall, there is no good logical flow in the writing. Many arguments seem to be non sequitur, e.g., the first and second paragraphs of 'Discussion". In the "Introduction", the fact that this has never been studied before is not a good enough argument for studying it. There must be better reasons, but they were not explained. Too many things have never been studied before because they may not be worth studying. 

Despite these serious concerns, I still think it's worth publishing these observations along with careful discussions of alternative interpretations.

---

## [Decision Letter · Decision Letter 2]

13 Jan 2023

Dear Dr Zhang,

Thank you for your patience while we considered your revised manuscript "N6-methyladenosine in monocyte-derived macrophages is involved in a mouse model of Alzheimer's disease" for publication as a Research Article at PLOS Biology. This revised version of your manuscript has been evaluated by the PLOS Biology editors, the Academic Editor and two of the original reviewers.

The reviews are appended below, and as you will see both Reviewers 1 and 3 are satisfied by the changes made in the revision and they suggest that we accept the manuscript for publication. While we are also satisfied by the changes made in response to reviewers, before we can accept your manuscript, we need you to address the following editorial requests: 

1) TITLE: After some discussion we think that the title should be edited slightly to add a bit more causality. If you agree, we would suggest you change it to something like "Loss of the m6A methyltransferase METTL3 in monocyte-derived macrophages ameliorates Alzheimer's disease pathology in mice"

2) ETHICS STATEMENT: Thank you for providing an ethics statement in the methods section of your paper. Please update this to include the approval number for the protocol approved by the IACUC of Xinxiang Medical University and the Animal Experiment Administration Committee of Fourth Military Medical University. Please also include the specific national or international regulations/guidelines to which your animal care and use protocol adhered. Please note that institutional or accreditation organization guidelines (such as AAALAC) do not meet this requirement.

3) BLURB: Please provide a blurb which (if accepted) will be included in our weekly and monthly Electronic Table of Contents, sent out to readers of PLOS Biology, and may be used to promote your article in social media. The blurb should be about 30-40 words long and is subject to editorial changes. It should, without exaggeration, entice people to read your manuscript. It should not be redundant with the title and should not contain acronyms or abbreviations

4) DATA REQUEST: Thank you for providing a supplemental file containing western blot images that accompany the bands presented in your main figures. Please note that we require the original, uncropped and minimally adjusted images supporting all blot and gel results reported in an article's figures or Supporting Information files. 

Some of the images provided in the current S1_raw_images file appear to be cropped (ex Fig 2B). Can you please provide an updated file containing the fully uncropped images in high resolution? Ideally, each membrane would be presented in its original size on its own page. 

Additionally, this file needs further annotations. Please indicate the loading order, identity of experimental samples, method used to capture the image for each image. Molecular weight markers should be included or indicated on the raw image, and any lanes not included in the final figure should be marked with an “X” above the lane label on the original blot/gel image. All labeling and annotation should be performed without obscuring any data or background bands.

For more information on our guidelines for how to prepare and upload this data see: https://journals.plos.org/plosbiology/s/figures#loc-blot-and-gel-reporting-requirements

---

We expect to receive your revised manuscript within two weeks. 

*Published Peer Review History*

*Press*

Sincerely,

Luke

Lucas Smith, Ph.D.

Associate Editor,

lsmith@plos.org,

PLOS Biology

Reviewer remarks:

Reviewer #1, Hua-Bing Li (note, Reviewer 1 has signed this review): The authors have made significant efforts to address all my concerns. I am satisified with the revision.

Reviewer #3: The authors have addressed all of my concerns. I believe it's suitable for publication in PLoS Biology.

---

## [Editor Report · Decision Letter 3]

30 Jan 2023

Dear Dr Zhang,

Thank you for the submission of your revised Research Article "Loss of the m6A methyltransferase METTL3 in monocyte-derived macrophages ameliorates Alzheimer's disease pathology in mice" for publication in PLOS Biology. We have now had a chance to assess the changes made in response to our last editorial requests and, on behalf of my colleagues and the Academic Editor, Richard Daneman, I am pleased to say that we can in principle accept your manuscript for publication, provided you address any remaining formatting and reporting issues. These will be detailed in an email you should receive within 2-3 business days from our colleagues in the journal operations team; no action is required from you until then. Please note that we will not be able to formally accept your manuscript and schedule it for publication until you have completed any requested changes.

**As a note: Thank you for providing me an updated supplementary file containing the uncropped western blot images accompanying your manuscript. I have gone ahead and replaced the previous version of that file with the new version that you provided me via email. Please do double check that everything looks good after this change. 

PRESS

Sincerely, 

Lucas Smith, Ph.D.

Associate Editor

PLOS Biology

lsmith@plos.org